# Endocrine Adverse Events in Patients Treated with Immune Checkpoint Inhibitors: A Comprehensive Analysis

**DOI:** 10.3390/medicina61010123

**Published:** 2025-01-14

**Authors:** Meriç Dökmetaş, Harun Muğlu, Erkan Özcan, Buket Bayram Kuvvet, Kaan Helvacı, Ender Kalacı, Seda Kahraman, Musa Barış Aykan, İrfan Çiçin, Fatih Selçukbiricik, Ömer Fatih Ölmez, Ahmet Bilici

**Affiliations:** 1Department of Medical Oncology, Faculty of Medicine, Medipol University, Istanbul 34810, Turkey; mericdokmetas@erciyes.edu.tr (M.D.); omerfatih.olmez@medipol.com.tr (Ö.F.Ö.); ahmet.bilici@medipol.com.tr (A.B.); 2Department of Medical Oncology, Faculty of Medicine, Trakya University, Edirne 22030, Turkey; erkan.ozcan1@saglik.gov.tr; 3Department of Medical Oncology, Faculty of Medicine, Koç University, Istanbul 34450, Turkey; bkuvvet@kuh.ku.edu.tr (B.B.K.); fselcukbiricik@kuh.ku.edu.tr (F.S.); 4Department of Medical Oncology, Memorial Hospital, Ankara 06520, Turkey; kaan.helvaci@memorial.com.tr; 5Department of Medical Oncology, Faculty of Medicine, Ankara University, Ankara 06620, Turkey; ender.kalaci@saglik.gov.tr; 6Department of Medical Oncology, Faculty of Medicine, Ankara Yıldırım Beyazıt University, Ankara Bilkent City Hospital, Ankara 06800, Turkey; seda.kahraman3@saglik.gov.tr; 7Department of Medical Oncology, Gülhane Training and Research Hospital, University of Health Sciences, Ankara 06010, Turkey; musabaris.aykan@saglik.gov.tr; 8Department of Medical Oncology, Faculty of Medicine, Istinye University, Istanbul 34010, Turkey; irfan.cicin@istinye.edu.tr

**Keywords:** immune checkpoint inhibitors (ICIs), immune-related adverse events (irAEs), endocrine dysfunction, cancer therapy

## Abstract

*Background and Objectives*: Immune checkpoint inhibitors (ICIs) have revolutionized cancer therapy, but their use is associated with a spectrum of immune-related adverse events (irAEs), including endocrine disorders. This study aims to investigate the incidence, timing, treatment modalities, and impact of ICI-related endocrine side effects in cancer patients. *Materials and Methods*: This retrospective study analyzed 139 cancer patients treated with ICIs between 2016 and 2022. Data regarding endocrine irAEs, including hypothyroidism, hyperthyroidism, hypophysitis, and diabetes mellitus, were collected. The study examined the timing of irAE onset, management approaches, and the association between irAEs and treatment outcomes. *Results*: The most common endocrine irAE was hypothyroidism (65.5%), followed by hyperthyroidism (2.3%), hypophysitis (8.6%), and diabetes mellitus (0.7%). These disorders typically emerged within the first six months of ICI therapy. Most cases were managed conservatively or with hormone replacement therapy. Patients who developed endocrine irAEs exhibited a higher objective response rate (ORR) and clinical benefit rate (CBR) compared to those without irAEs. *Conclusions*: Endocrine dysfunction is a significant toxicity of ICI therapy. Early recognition, prompt diagnosis, and appropriate management are crucial to minimize their impact on patient health and quality of life. This study highlights the potential association between irAEs and improved clinical outcomes. Further research is needed to elucidate the underlying mechanisms and identify predictive biomarkers for irAE development.

## 1. Introduction

Immune checkpoint inhibitors (ICIs) have revolutionized cancer therapy by harnessing the body’s own immune system to combat malignant cells. The immune system is a defense mechanism that recognizes and neutralizes organisms and molecules in our body. Its relationship with cancer has been a research focus since the 19th century, particularly in tumor immunology studies. These agents, targeting key immune checkpoints such as programmed cell death receptor 1 (PD-1), programmed cell death ligand 1 (PD-L1), and cytotoxic T lymphocyte-associated antigen 4 (CTLA-4), have demonstrated significant clinical efficacy across various malignancies. ICIs include drugs such as PD-1 inhibitors (nivolumab, pembrolizumab, cemiplimab, dostarlimab), PD-L1 inhibitors (atezolizumab, avelumab, durvalumab), and CTLA-4 inhibitors (ipilimumab, tremelimumab), many of which are FDA-approved for treating different cancers. However, their therapeutic potential is tempered by a spectrum of immune-related adverse events (irAEs), including endocrinological disorders [1,2,3].

Endocrine dysfunction, a less recognized but increasingly prevalent irAE, can manifest as hypothyroidism, hyperthyroidism, hypophysitis, adrenal insufficiency (AI), and diabetes mellitus (DM). These conditions are often difficult to diagnose because they present nonspecific symptoms. Unlike other irAEs, immunosuppressive agents are generally not used in their treatment, and hormone replacement therapy is typically administered. These conditions, often presented with subtle and nonspecific symptoms, can significantly impact patients’ quality of life and overall prognosis. Rapid recognition and management of endocrinological irAEs are crucial, as they can have serious consequences, including impacts on cancer therapy. Prompt recognition and appropriate management are crucial to mitigating their adverse consequences [3,4,5].

Immune checkpoint inhibitors (ICIs) can cause a variety of endocrine-related adverse effects, each with unique pathophysiological mechanisms. Thyroid-related adverse events are often associated with the presence of thyroid autoantibodies, such as anti-thyroglobulin (TG) and anti-thyroid peroxidase (TPO), prior to ICI therapy. Patients with positive baseline antibodies have a significantly higher risk of developing thyroid dysfunction compared to those without these antibodies. Elevated cytokine levels, such as IL-1β and IL-2, have also been linked to thyroid dysfunction following PD-1 inhibitor therapy. Interestingly, thyroid-related events have been associated with improved overall survival in cancer patients, although the underlying mechanisms remain unclear [6].

Hypophysitis, most seen with anti-CTLA-4 therapy, is thought to result from a combination of type II hypersensitivity reactions involving IgG or IgM antibodies and T cell-mediated type IV hypersensitivity. The presence of specific anti-pituitary antibodies, such as anti-thyrotroph, anti-corticotroph, and anti-gonadotroph antibodies, plays a central role in the development of this condition. Additionally, the expression of CTLA-4 in pituitary tissues may contribute to the higher incidence of hypophysitis with this type of therapy [7].

ICI-induced primary adrenal insufficiency (ICI-PAI) occurs due to T cell and antibody-mediated destruction of the adrenal cortex. Imaging studies have demonstrated increased 18F-fluorodeoxyglucose uptake in the adrenal glands of patients with ICI-PAI, supporting the hypothesis of an inflammatory process as the underlying mechanism. Patients often present with adrenal insufficiency symptoms, and the condition requires prompt recognition and management [8].

ICI-induced diabetes mellitus (ICI-DM) is primarily driven by the autoimmune destruction of pancreatic β-cells, facilitated by the inhibition of the PD-1 and PD-L1 pathway. This destruction leads to the loss of insulin-producing cells, which is more severe and rapid compared to classical type 1 diabetes mellitus (T1DM). Interestingly, only 40–50% of ICI-DM patients have detectable islet autoantibodies at diagnosis, compared to over 90% in T1DM. Genetic factors, particularly certain HLA haplotypes like DR3-DQ2 and DR4-DQ8, have been implicated in increasing the risk of ICI-DM. Inflammation of the pancreas can also lead to exocrine dysfunction, with elevated lipase and amylase levels frequently observed [9].

ICI-induced hypoparathyroidism is rare, and its exact mechanism is not well understood. Proposed mechanisms include autoantibody-induced cell damage of the parathyroid glands or inhibition of parathyroid hormone secretion through calcium-sensing receptor autoantibodies. Collectively, these endocrine adverse effects highlight the diverse and complex pathophysiological impacts of ICIs, necessitating careful monitoring and tailored management strategies [10].

To address this emerging clinical challenge, our study retrospectively analysed a cohort of 516 solid organ cancer patients treated with ICIs between 2016 and 2022. We aimed to investigate the incidence, timing, treatment modalities, and impact of ICI-related endocrinological side effects on cancer therapy and patient survival. By elucidating the clinical characteristics and management strategies for these disorders, we hope to improve patient outcomes and optimize the therapeutic potential of ICIs.

## 2. Materials and Methods

### 2.1. Study Design

This multicenter retrospective study analysed 139 patients with solid organ cancers who developed immune checkpoint inhibitor (ICI)-related endocrinological side effects between 2016 and 2022. Patient data including demographics, cancer characteristics, ICI treatment details, and endocrine side effect profiles were collected. The participating centers in this study included Istanbul Medipol University Faculty of Medicine (Istanbul, Turkey), Ankara University Faculty of Medicine (Ankara, Turkey), Trakya University Faculty of Medicine (Edirne, Turkey), Koç University Faculty of Medicine (Istanbul, Turkey), Ankara Yıldırım Beyazıt University Faculty of Medicine (Ankara, Turkey), Yüksek Ihtisas University Faculty of Medicine (Ankara, Turkey), and Gülhane Training and Research Hospital (Ankara, Turkey). This study included patients aged 18 or older with solid organ cancers treated with immune checkpoint inhibitors (ICIs) between 2016 and 2022, who developed endocrine-related adverse events (e.g., hypothyroidism, hyperthyroidism, hypophysitis, or type 1 diabetes mellitus). Inclusion required complete medical records, diagnostic confirmation (e.g., lab tests, imaging), and ethical approval.

Patients with non-endocrine adverse events, incomplete data, insufficient follow-up (<3 months), or unconventional treatments were excluded. Pregnant or breastfeeding patients were also excluded to ensure a focused and reliable analysis.

The study defined and characterized ICI-associated hypothyroidism, hyperthyroidism, hypophysitis, and type 1 diabetes mellitus (T1DM). Diagnostic criteria, treatment strategies, and response assessments for these conditions were outlined. Thyroid function tests, cortisol levels, pituitary hormone levels, serum glucose values, thyroid ultrasonography (USG) for thyroid side effects, and pituitary magnetic resonance imaging (MRI) for hypophysitis were evaluated retrospectively. The impact of these endocrine side effects on overall survival was also evaluated. Grade determination of ICI-related side effects was performed using the ASCO “Management of Immune-Related Adverse Events in Patients Treated with Immune Checkpoint Inhibitor Therapy” guideline [5]. Patients who did not have a known thyroid disease or thyroid dysfunction at baseline, and whose serum fT4 level was low and serum TSH level was measured during ICI treatment or during follow-up after the end of treatment, were defined as ICI-associated hypothyroidism. Patients with suppressed TSH and high fT4 and fT3 levels were defined as ICI-related hyperthyroidism. Among these patients, information about those with thyroid autoantibodies and thyroid USG was recorded. In addition, deteriorations in thyroid functions were also considered as ICI-related side effects in patients with known thyroid disorders who received treatment or who were asymptomatic and did not receive treatment.

ICI-associated hypophysitis was defined in patients with a basal serum cortisol level < 5 mcg/dL and an ACTH level of 20 pg/mL, based on the presence of a peak cortisol response in the insulin tolerance or ACTH stimulation test when the basal cortisol level was <18 mcg/dL. This was accompanied by low serum IGF-1 and fT4 levels, low or normal TSH levels, suppressed FSH and LH levels, and radiological evidence of hypophysitis on pituitary MRI.

ICI-associated T1DM was defined as high serum glucose levels, an insulin requirement, low serum c-peptide levels, and positive T1DM-related antibodies in patients taking ICIs with or without previously known DM.

Before the research, approval was received from the Ethics Committee of Istanbul Medipol University Faculty of Medicine (Document number: E-10840098-772.02-6952, date: 15 November 2022).

### 2.2. Baseline Characteristics of Patients

This study analyzed 139 patients with various solid organ cancers who received ICI therapy. The majority of patients were male (68.3%) with a median age of 62 years. Approximately 42.4% of patients were under 60 years old, and a significant portion (59%) had no comorbidities. Common comorbidities included hypertension, diabetes, and cardiovascular disease. Cancer staging was performed using the 8th version of the AJCC/UICC TNM staging system based on clinical and radiological findings. Endocrinological side effects were graded according to the ASCO “Management of Immune-Related Adverse Events in Patients Treated with Immune Checkpoint Inhibitor Therapy” guideline [5].

Regarding cancer types, non-small cell lung cancer (NSCLC) was the most prevalent (47.5%), followed by renal cell carcinoma, malignant melanoma, and small cell lung cancer. A significant number of patients (80.6%) were diagnosed at stage 4, indicating an advanced disease.

In terms of treatment lines, 24.5% of patients received ICI as a first-line therapy, while the majority received it as second-line or later-line treatment. Pembrolizumab, nivolumab, and atezolizumab were the most commonly used ICI agents (Table 1).

### 2.3. Statistical Analysis

All statistical analyses were performed using SPSS version 24.0. Descriptive statistics for continuous variables were presented as medians with interquartile ranges or 95% confidence intervals, as appropriate. Categorical variables were summarized as frequencies and percentages.

To assess relationships between the immune checkpoint inhibitor (ICI) treatment response and clinicopathological factors, as well as treatment preferences, chi-square and Fisher’s exact tests were employed. Kaplan–Meier analysis was used to construct survival curves, with the log-rank test applied to compare differences between groups. Overall survival was defined as the time from diagnosis to death or loss to follow-up.

Univariate analysis was initially performed to identify potential prognostic factors. Variables with a *p*-value < 0.10 in univariate analysis were included in multivariate analysis using the Cox-proportional-hazards regression model. The proportional hazards assumption was tested for all Cox models to ensure the validity of the survival analysis.

Logistic regression analysis was conducted to identify independent predictors of the response to ICI treatment. Both univariate and multivariate models were applied, with variables selected based on clinical relevance and statistical significance. Odds ratios (ORs) and their 95% confidence intervals (CIs) were reported to quantify the strength of associations.

The relationships between overall survival and endocrinological adverse events (irAEs) were also evaluated. For this, a subgroup analysis was performed to explore differences in outcomes between patients with and without endocrine irAEs. Statistical significance was set at *p* < 0.05 for all analyses, with *p*-values reported as two-sided to account for the possibility of effects in both directions.

To ensure the robustness of findings, additional post-hoc power analyses were conducted to confirm the adequacy of the sample size for detecting meaningful differences in survival and response rates. Graphical visualizations, including Kaplan–Meier curves and forest plots, were employed to enhance the clarity of results and to facilitate interpretation by the readers.

## 3. Results

### 3.1. Incidence and Spectrum of Endocrine irAEs

This study analysed 139 patients with solid organ cancer who received ICI therapy (alone or in combination). A total of eighty percent of the patients included in the study had metastatic disease, seventeen percent had locally advanced disease, and two percent were in the early stage. The most common endocrine adverse event was hypothyroidism (65.5%), while the least common was diabetes mellitus (0.7%). No patient developed hypoparathyroidism. Other adverse events included hyperthyroidism in 2.3% of patients, autoimmune disease in 13.7% of patients, and pituitaryitis in 8.6% of patients. Most adverse events were mild (grade 1 or 2) according to the ASCO grading system. Of the patients who developed hypothyroidism, 35.2% experienced grade 1 adverse events, 61.5% grade 2 adverse events, 2.2% grade 3 adverse events, and 1.1% grade 4 adverse events. Among patients who developed hyperthyroidism, 51.6% experienced grade 1 adverse events, and 48.6% grade 2 adverse events, with no grade 3 or 4 events reported. For patients with pituitaryitis, 25% experienced grade 1 adverse events, 58.3% grade 2 adverse events, and 16.7% grade 3 adverse events, with no grade 4 events. Of patients who developed autoimmune disease, 31.5% experienced grade 1 adverse events, 42.1% grade 2 adverse events, 15.7% grade 3 adverse events, and 10.7% grade 4 adverse events. A single patient developed grade 4 diabetes mellitus (Table 2).

### 3.2. Timing of irAE Onset

Immunotherapy induced side effects emerged with a median latency of 4.5 months (range: 0.4–20.5 months). Notably, 29.5% of patients developed adverse events after six months of treatment (Figure 1).

### 3.3. Management of Endocrine irAEs

Among the 91 patients who developed hypothyroidism, 65.5% were managed conservatively, while 39.1% required hormone replacement therapy. In two cases, immunotherapy was temporarily paused, and in another two, it was discontinued. A severe case of myxedema coma occurred in a 69-year-old patient, necessitating intensive treatment. A total of one patient with pre-existing hypothyroidism experienced worsening symptoms, and two patients with subclinical hypothyroidism progressed to overt disease. TSH levels should be monitored 6–8 weeks after treatment initiation to achieve stable levels, and levothyroxine should be titrated to normalize TSH levels. In central hypothyroidism, free T4 should be monitored instead of TSH. Additionally, adrenal insufficiency should be ruled out and treated prior to initiating thyroid hormone replacement to prevent adrenal crisis [11].

Routine monitoring for secondary adrenal insufficiency in ICI-treated patients is controversial. Morning cortisol levels with or without ACTH may be obtained in patients receiving anti-CTLA-4 inhibitors or combination therapy but are not typically recommended to be obtained for anti-PD-1/PD-L1 monotherapy without symptoms. Adrenal insufficiency is treated with glucocorticoid replacement, such as hydrocortisone or prednisone, with education on sick day dosing to prevent adrenal crisis. Stress dosing and hospitalization for IV hydration or parenteral glucocorticoid therapy are necessary for severe illness or for procedures. If central hypothyroidism is diagnosed, glucocorticoids should be administered before thyroid hormone replacement to avoid adrenal crisis. High-dose glucocorticoids may be required for severe hypophysitis-related symptoms like headaches and vision changes. Immunotherapy can generally continue with proper hormone replacement therapy once severe symptoms resolve [12].

Primary adrenal insufficiency (PAI), if untreated, can be life-threatening and requires prompt glucocorticoid replacement therapy. The dose and route should be based on clinical status, with lifelong glucocorticoid and mineralocorticoid replacement therapy required to prevent hypotension and hyperkalemia. ICI therapy does not need to be discontinued unless symptoms are severe [13].

ICI-related diabetes mellitus (ICI-DM) often presents as diabetic ketoacidosis (DKA) in 60–85% of cases. Immediate treatment follows standard DKA protocols, including intravenous insulin, fluid resuscitation, and the correction of electrolyte abnormalities. Patients require permanent insulin therapy, managed similarly to classic type 1 diabetes. High-dose glucocorticoids are not recommended as they do not improve outcomes and may exacerbate hyperglycemia. Experimental therapies, such as mesenchymal stem cell treatment, have shown potential in preclinical models, but further research is needed [14,15].

Of the 31 patients with hyperthyroidism, 4.3% received antithyroid medication, while the majority were monitored. Most subsequently developed hypothyroidism. Immunotherapy was temporarily paused in five cases, but no treatment discontinuation was required. Treatment of ICI-induced thyrotoxicosis due to destructive thyroiditis is mainly supportive, with beta-blockers used for symptom control. In cases of ICI-induced Graves’ disease, anti-thyroidal medications, radioactive iodine, or surgery may be considered based on patient preference and clinical presentation. High-dose glucocorticoids are not recommended for thyroid-related immune-related endocrine events (irEEs) as they do not alter disease progression.

A total of 91 patients developed hypothyroidism, with thyroid autoantibodies detected in 31.9%. Among these, 34.4% were anti-TPO positive, 13.7% were anti-Tg positive, 17.2% were positive for both, and 27.5% were negative for both. Thyroiditis was observed in 86.2% of patients with available thyroid ultrasound data. A total of 31 patients developed hyperthyroidism. Thyroid autoantibodies were detected in 58% of these patients, primarily anti-TPO. Thyroiditis was present in 75% of patients with available ultrasound data.

Hypophysitis occurred in 12 patients, all of whom required hormone replacement therapy. Of these patients, one patient received high-dose steroids. Immunotherapy was temporarily paused in seven patients. Diabetes mellitus developed in one patient, who was treated with insulin therapy and had a temporary pause in immunotherapy. Adrenal insufficiency occurred in 19 patients, all of whom received hormone replacement therapy. A total of nine patients received high-dose corticosteroids. Immunotherapy was paused in two patients and discontinued in three.

For male hypogonadotropic hypogonadism, testosterone replacement can be initiated if indicated and there is no contraindication. In cases of ICI-CDI, desmopressin is commonly utilized as an effective therapeutic option to manage severe thirst and polyuria, which are hallmark symptoms of the condition. For patients experiencing severe dehydration and hypernatremia, temporary suspension of ICI therapy may be warranted to prevent further complications. This suspension should be maintained until the patient achieves adequate clinical stabilization, after which resumption of ICI therapy can be carefully considered based on the overall clinical status and the risk-benefit assessment.

### 3.4. Impact of Endocrine irAEs on Clinical Outcomes

Of the 122 patients, 25 achieved a complete response (CR), 49 a partial response (PR), 29 stable disease (SD), and 28 progressive diseases (PD). The objective response rate (ORR) was 53.3%, and the clinical benefit rate (CBR) was 74.2%. Male gender and treatment with pembrolizumab were independent predictors of response to immunotherapy. Endocrine side effects did not significantly impact response rates. The median overall survival was 40.9 months (Figure 2). No deaths were attributed to endocrine side effects.

## 4. Discussion

Endocrine dysfunction is a significant toxicity associated with ICI therapy, impacting up to 40% of treated patients. The most affected organs are the thyroid gland, the pituitary gland, the adrenal glands, and the pancreas. Thyroid dysfunction, often manifesting as hypothyroidism preceded by transient hyperthyroidism, is the most prevalent endocrine side effect. Other potential complications include panhypopituitarism, primary adrenal insufficiency, and insulin-deficient diabetes. These endocrine disorders are distinct from those caused by conventional chemotherapy or newer targeted therapies [16].

ICI-induced endocrinopathies are a significant side effect of immune checkpoint inhibitor therapy. These endocrine disorders, such as hypothyroidism and diabetes mellitus, can vary widely in severity and the timing of onset. While mild cases of hypothyroidism are common, severe cases of diabetes mellitus can be life-threatening. Early detection and timely intervention with hormone replacement therapy are crucial for managing these conditions and improving the patients’ quality of life. Given the subtle and non-specific nature of symptoms, oncologists must maintain a high index of suspicion and implement regular screening protocols to identify and address these endocrine disorders promptly [17].

Thyroid dysfunction is a common side effect of immune checkpoint inhibitor (ICI) therapy, affecting approximately 10% of patients on anti-PD-1/PD-L1 monotherapy and up to 20% of those on combination therapy with CTLA-4 inhibitors. Many cases manifest hypothyroidism, often preceded by destructive thyroiditis. While pre-existing anti-thyroid antibodies increase the risk, the exact mechanisms underlying ICI-induced thyroid dysfunction are not fully understood. Hypothyroidism typically presents subtle symptoms like fatigue, weight gain, and cognitive changes. Thyroiditis can lead to transient hyperthyroidism, but severe cases are rare. Diagnosis relies on laboratory tests, primarily of TSH and FT4 levels. In cases of secondary hypothyroidism, further evaluation of pituitary function is necessary. Regular monitoring of thyroid function is crucial. Hypothyroidism is treated with levothyroxine replacement therapy. For thyrotoxicosis, supportive care and beta-blockers may be sufficient. In rare cases of severe thyrotoxicosis, additional treatments like anti-thyroid medications or radioactive iodine may be considered [7,17,18,19,20,21,22,23].

Hypophysitis, an inflammation of the pituitary gland, is a rare complication of immune checkpoint inhibitor (ICI) therapy, primarily associated with anti-CTLA-4 based treatments. This condition can lead to hypopituitarism, a deficiency in one or more of the pituitary hormones. The exact mechanism is not fully understood, but it is believed to involve a type II hypersensitivity reaction. Autoantibodies targeting specific pituitary cells may play a role in this process. Patients often present with non-specific symptoms such as fatigue, nausea, and headaches. More severe symptoms may include visual disturbances, hypotension, and adrenal crisis. Laboratory findings may reveal low levels of cortisol, thyroid hormone, and sex hormones. Diagnosis typically involves a combination of clinical presentation, laboratory tests, and imaging studies. A brain MRI can reveal pituitary enlargement or an empty sella. Treatment focuses on hormone replacement therapy to address deficiencies. Glucocorticoid replacement is essential for adrenal insufficiency, while thyroid hormone and sex hormone replacement may also be required. In severe cases, high-dose corticosteroids may be used to reduce inflammation. It is important to note that while ICI therapy can be continued in many cases, careful monitoring and dose adjustments may be necessary [24,25,26].

Primary adrenal insufficiency is a relatively uncommon but potentially life-threatening complication associated with immune checkpoint inhibitor (ICI) therapy. This condition arises when the adrenal glands, responsible for producing crucial hormones like cortisol and aldosterone, are compromised. Patients with primary adrenal insufficiency may experience symptoms such as fatigue, malaise, nausea, hypotension, and in severe cases, adrenal crisis. These symptoms stem from the combined deficiency of glucocorticoids and mineralocorticoids. Laboratory tests typically reveal low morning cortisol levels and elevated ACTH levels. Additionally, metabolic acidosis and hyperkalaemia may be present if mineralocorticoid-producing cells are affected. Given the serious nature of primary adrenal insufficiency, immediate treatment is essential. This involves initiating glucocorticoid replacement therapy, like the approach for secondary adrenal insufficiency. In some cases, mineralocorticoid supplementation may also be required. While ICI therapy may be continued following acute stabilization, close monitoring and appropriate management of adrenal insufficiency are crucial [27].

Immune checkpoint inhibitor (ICI)-associated diabetes mellitus (ICI-DM) is a serious side effect affecting approximately 1% of patients treated with ICIs. It is characterized by a severe and persistent insulin deficiency, often leading to diabetic ketoacidosis (DKA). The underlying mechanism involves immune-mediated destruction of pancreatic beta cells, likely triggered by an increased PD-L1 expression on these cells. While sharing similarities with type 1 diabetes, ICI-DM exhibits distinct features, including a more rapid onset and a lower prevalence of islet autoantibodies. Genetic factors, particularly HLA haplotypes associated with type 1 diabetes, also play a role in susceptibility to ICI-DM. ICI-DM can develop at any time during or after ICI therapy, with a median onset of 7–17 weeks. Symptoms include polyuria, polydipsia, fatigue, and abdominal pain. Diagnosis often relies on random blood glucose measurements and basic metabolic panels, as A1C levels may not be significantly elevated in acute cases. Regular glucose monitoring is essential for the early detection of ICI-DM. Treatment primarily involves insulin therapy to manage blood glucose levels. Unlike other ICI-related endocrinopathies, high-dose steroids are not indicated for ICI-DM. While some cases of remission have been reported, most patients require lifelong insulin therapy [28,29,30,31,32,33,34,35].

While less common, immune checkpoint inhibitor (ICI) therapy has also been associated with the development of more rare endocrine disorders. These include primary hypoparathyroidism, diabetes insipidus, syndrome of inappropriate anti-diuretic hormone secretion (SIADH), and Cushing’s disease. Due to the extremely low incidence of these conditions, detailed information is limited. Diagnosis and management of these rare endocrine complications should rely on clinical presentation and established treatment guidelines.

Our study further highlights the association between irAEs and improved clinical outcomes. Patients who developed endocrine irAEs had a higher objective response rate and clinical benefit rate compared to those without irAEs. This suggests that irAEs may serve as a biomarker for effective immunotherapy response. Nutrients play a pivotal role in modulating immune function and maintaining immune homeostasis. Macronutrients, such as proteins, carbohydrates, and fatty acids, along with micronutrients like vitamins, minerals, antioxidants, and probiotics, influence both innate and adaptive immunity. They regulate inflammation and cytokine expression, modulate immune cell signaling, and impact T and B cell activation, differentiation, and antibody production. These interactions underscore the importance of optimizing nutritional strategies to support immune function and potentially mitigate immune-related adverse events in patients undergoing immune checkpoint inhibitor therapy [36].

However, it is important to note that the development of irAEs can lead to significant morbidity and may require a dose reduction or treatment interruption. Therefore, a careful balance must be struck between maximizing the therapeutic benefit and minimizing adverse effects [37].

Our study has several limitations, including its retrospective design and relatively small sample size. Further prospective studies with larger cohorts are needed to validate our findings and to investigate the underlying mechanisms of ICI-induced endocrine disorders. Endocrine irAEs are common complications of ICI therapy. Recent studies have found that high serum levels of selenium and zinc are associated with a better prognosis in cancer patients, while elevated copper levels are linked to poorer survival outcomes. These associations suggest that serum levels of certain elements may serve as prognostic markers across various cancers [38]. Early recognition, prompt diagnosis, and appropriate management are essential to minimize their impact on patient health and quality of life. Future research should focus on identifying predictive biomarkers for irAE development and developing strategies to mitigate their severity.

## 5. Conclusions

Immune checkpoint inhibitors (ICIs) have revolutionized cancer therapy, but their use is accompanied by a range of immune-related adverse events (irAEs), including endocrine disorders. Our retrospective study highlights the significant prevalence of ICI-associated endocrine dysfunction, particularly hypothyroidism, hyperthyroidism, hypophysitis, and diabetes mellitus. We found that early detection and timely management of these endocrine disorders are crucial for optimizing patient outcomes. Hormone replacement therapy remains the cornerstone of treatment, while in some cases, temporary or permanent discontinuation of ICI therapy may be necessary.

Furthermore, our study suggests a potential association between the development of irAEs and improved clinical outcomes. This intriguing finding warrants further investigation to elucidate the underlying mechanisms and to determine whether irAEs can serve as biomarkers for an effective immunotherapy response. While ICI therapy has significantly advanced cancer treatment, it is imperative to carefully monitor patients for the development of endocrine irAEs. A multidisciplinary approach involving oncologists, endocrinologists, and other specialists is essential to ensure optimal management of these complications.

Future research should focus on identifying predictive biomarkers for irAE development, developing strategies to mitigate their severity, and improving our understanding of the underlying immunological mechanisms.

## Figures and Tables

**Figure 1 medicina-61-00123-f001:**
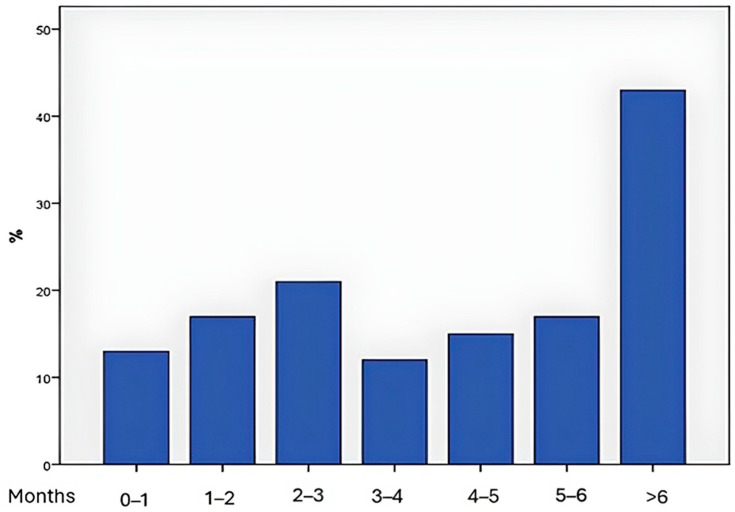
Timeline of endocrinologic side effects, time(month).

**Figure 2 medicina-61-00123-f002:**
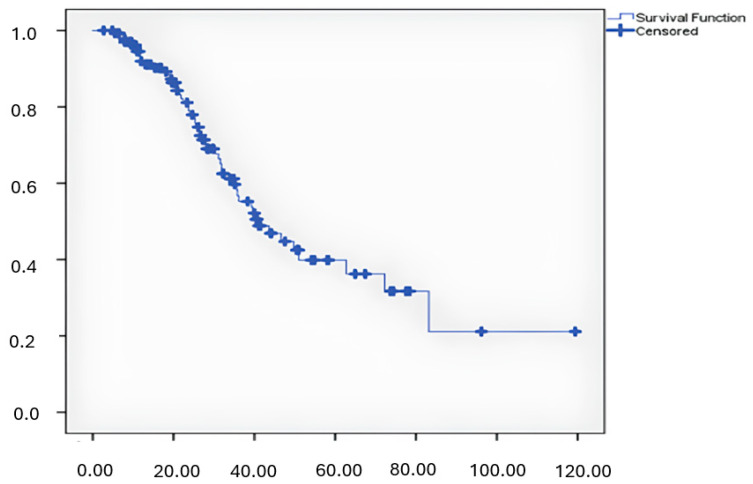
Overall survival, time(month).

**Table 1 medicina-61-00123-t001:** Baseline characteristics.

Variables	No. (%)
Sex—no. (%)	
Female	44 (31.7)
Male	95 (68.3)
Age—no. (%)	
Median(range)-year	62 (24–84)
<60-year	59 (42.4)
>60-year	80 (57.6)
Comorbidity—no. (%)	
Yes	57 (41.0)
No	82 (59.0)
Solid Tumor Type—no. (%)	
NSCLC *	66 (47.5)
SCLC ^†^	11 (7.9)
Malign Melanoma	16 (11.5)
RCC ^‡^	18 (12.9)
Bladder cancer	3 (2.2)
Gastric cancer	5 (3.6)
HCC ^§^	4 (2.9)
CRC ^a^	4 (2.9)
Head and Neck cancer	3 (2.2)
Pleural Mesothelioma	3 (2.2)
Breast Cancer	1 (0.7)
Endometrium cancer	2 (1.4)
Prostate cancer	1 (0.7)
Sarcoma	2 (1.4)
TNM Stage in Diagnosis—no. (%)	
2	3 (2.2)
3	24 (17.3)
4	112 (80.6)
Number of treatments prior to ICI ^b^—no. (%)	
0	34 (24.5)
1	49 (35.3)
2	17 (12.2)
3	8 (5.8)
≥4	21 (15.1)
Unknown	10 (7.1)
ICI ^b^ type	
Nivolumab	40 (28.8)
Pembrolizumab	52 (37.4)
Atezolizumab	24 (17.3)
İpilimumab	1 (0.7)
Durvalumab	9 (6.5)
Cemiplimab	3 (2.2)
İpilimumab + Nivolumab	6 (4.3)
Pembrolizumab + İpilimumab	3 (2.2)
Serplulimab	1 (0.7)

* NSCLC: non-small cell lung cancer; ^†^ SCLC: small cell lung cancer; ^‡^ RCC: renal cell carcinoma; ^§^ HCC: hepatocellular carcinoma; ^a^ CRC: colorectal carcinoma; ^b^ ICI: immune checkpoint inhibitors.

**Table 2 medicina-61-00123-t002:** Frequency and severity of ICI ^‡^-related side effects.

Side Effect	*n* (%)	Grade 1 *n* (%)	Grade 2 *n* (%)	Grade 3 *n* (%)	Grade 4 *n* (%)
Hypothyroidism	91 (65.5)	32 (35.2)	56 (61.5)	2 (2.2)	1 (1.1)
Hyperthyroidism	31 (22.3)	16 (51.6)	15 (48.6)	0	0
Hypophysitis	12 (8.6)	3 (25.0)	7 (58.3)	2 (16.7)	0
DM ^†^	1 (0.7)	0	0	0	1 (100)
AI *	19 (13.7)	6 (31.5)	8 (42.1)	3 (15.7)	2 (10.7)

* AI: adrenal insufficiency; ^†^ DM: diabetes mellitus; ^‡^ ICI: immune checkpoint inhibitors.

## Data Availability

Data are contained within the article.

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
