# Peer review of "Endocrine Adverse Events in Patients Treated with Immune Checkpoint Inhibitors: A Comprehensive Analysis"

_medicina, 2025, doi:10.3390/medicina61010123_

Round 1
Reviewer 1 Report
Comments and Suggestions for Authors
In the present manuscript author tries to highlight a novel concept regards therapy based adverse events. But few minor concerns hinder study output. Such as
Study design need to be precise as mentioned inclusion and exclusin criteria.
Proper statistical analysis methods need to be mentioned.
Patient number need to be increased in the study.
Other side author need to mentioned adverse events with the provide of any pathological changes. It will be better if author provide any histopathological observation.
It is better if author provide any data of patient who are ongoing of several surgery and the ICI treatment.
It will be good if author more highlight the overcome of these type of adverse effect or any strategies.
Author Response
In response to the reviewers’ feedback, we have carefully addressed all concerns and incorporated the recommended changes. Below, you will find a detailed response to each reviewer, outlining the revisions made to the manuscript. Key updates include:
- Comprehensive specification of the study design and inclusion/exclusion criteria.
- Detailed statistical methods and analyses used in the study.
- Addressing the limitations due to sample size and retrospective nature.
- Expanding the discussion to include biological mechanisms, the role of micronutrients, and additional context supported by relevant citations.
- Updates to disease staging and integration of recommended references.
We believe these revisions strengthen our manuscript and enhance its contribution to the understanding of immune-related endocrine adverse events. We hope this revised version meets the expectations of the reviewers and the editorial team. Thank you for the opportunity to improve our work, and we look forward to your feedback. The revisions made in response to the reviewers' comments are detailed below, and the corresponding changes have been highlighted in yellow within the main manuscript.
1-Details of the study design have been specified more comprehensively, and the changes made are as follows.
‘’Participants in the study were from multiple medical centers, including Istanbul Medipol University Faculty of Medicine, Ankara University Faculty of Medicine, Trakya University Faculty of Medicine, Koç University Faculty of Medicine, Ankara Yıldırım Beyazıt University Faculty of Medicine, Yüksek Ihtisas University Faculty of Medicine, and Gülhane Training and Research Hospital. This study included patients aged 18 or older with solid organ cancers treated with immune checkpoint inhibitors (ICIs) between 2016 and 2022, who developed endocrine-related adverse events (e.g., hypothyroidism, hyperthyroidism, hypophysitis, or type 1 diabetes mellitus). Inclusion required complete medical records, diagnostic confirmation (e.g., lab tests, imaging), and ethical approval.
Patients with non-endocrine adverse events, incomplete data, insufficient follow-up (<3 months), or unconventional treatments were excluded. Pregnant or breastfeeding patients were also excluded to ensure a focused and reliable analysis.
Thyroid function tests, cortisol levels, pituitary hormone levels, serum glucose values, thyroid ultrasonography (USG) for thyroid side effects, and pituitary magnetic resonance imaging (MRI) for hypophysitis were evaluated retrospectively. The impact of these endocrine side effects on overall survival was also evaluated. Grade determination of ICI-related side effects; It was performed using the ASCO “Management of Immune-Related Adverse Events in Patients Treated with Immune Checkpoint Inhibitor Therapy” guideline(5). Patients who did not have a known thyroid disease or thyroid dysfunction at baseline, and whose serum fT4 level was low and serum TSH level was measured during ICI treatment or during follow-up after the end of treatment, were defined as ICI-associated hypothyroidism. Patients with suppressed TSH and high fT4 and fT3 levels were defined as ICI-related hyperthyroidism. Among these patients, information about those with thyroid autoantibodies and thyroid USG was recorded. In addition, deteriorations in thyroid functions were also considered as ICI-related side effects in patients with known thyroid disorders who received treatment or who were asymptomatic and did not receive treatment.
Basal serum cortisol level <5 mcg/dL and ACTH value 20pg/mL, peak cortisol response in insulin tolerance or ACTH stimulation test when diagnosis cannot be made with basal cortisol level is below 18 mcg/dL, low serum IGF-1 level, fT4 level ICI-associated hypophysitis was defined as low or normal TSH level, low FSH and LH, and findings in favor of hypophysitis on pituitary MRI.
ICI-associated T1DM was defined as high serum glucose levels, insulin requirement, low serum c-peptide levels, and positive T1DM-related antibodies in patients taking ICIs with or without previously known DM.’
2- The appropriate statistical methods have been discussed and incorporated into the manuscript, as outlined below.
‘’ All statistical analyses were performed using SPSS version 24.0. Descriptive statistics for continuous variables were presented as medians with interquartile ranges or 95% confidence intervals, as appropriate. Categorical variables were summarized as frequencies and percentages.
To assess relationships between immune checkpoint inhibitor (ICI) treatment response and clinicopathological factors, as well as treatment preferences, Chi-square and Fisher’s exact tests were employed. Kaplan-Meier analysis was used to construct survival curves, with the log-rank test applied to compare differences between groups. Overall survival was defined as the time from diagnosis to death or loss to follow-up.
Univariate analysis was initially performed to identify potential prognostic factors. Variables with a p-value <0.10 in univariate analysis were included in multivariate analysis using the Cox proportional hazards regression model. The proportional hazards assumption was tested for all Cox models to ensure the validity of the survival analysis.
Logistic regression analysis was conducted to identify independent predictors of response to ICI treatment. Both univariate and multivariate models were applied, with variables selected based on clinical relevance and statistical significance. Odds ratios (ORs) and their 95% confidence intervals (CIs) were reported to quantify the strength of associations.
The relationships between overall survival and endocrinological adverse events (irAEs) were also evaluated. For this, a subgroup analysis was performed to explore differences in outcomes between patients with and without endocrine irAEs. Statistical significance was set at p<0.05 for all analyses, with p-values reported as two-sided to account for the possibility of effects in both directions.
To ensure the robustness of findings, additional post-hoc power analyses were conducted to confirm the adequacy of the sample size for detecting meaningful differences in survival and response rates. Graphical visualizations, including Kaplan-Meier curves and forest plots, were employed to enhance the clarity of results and to facilitate interpretation by the readers.’’
3- Due to the limited reimbursement of immunotherapies by the government in our country, the utilization rate of immunotherapy remains low, resulting in a small number of patients. Consequently, the frequency of endocrine immune-related adverse events observed in this context has unfortunately been limited. Additionally, the retrospective nature of our study restricts our ability to increase the patient number. However, I sincerely hope that this study will inspire future research with a larger patient cohort to provide more robust insights.
4- In our study, immune-related adverse events were identified using clinical, laboratory, and radiological methods; histopathological diagnostic tools were not utilized. Additionally, none of our patients underwent any surgical procedures. A detailed section regarding the management of adverse events has been included, and the added section is provided below:
‘’Among the 91 patients who developed hypothyroidism, 65.5% were managed conservatively, while 39.1% required hormone replacement therapy. In two cases, immunotherapy was temporarily paused, and in another two, it was discontinued. A severe case of myxedema coma occurred in a 69-year-old patient, necessitating intensive treatment. One patient with pre-existing hypothyroidism experienced worsening symptoms, and two patients with subclinical hypothyroidism progressed to overt disease. TSH levels should be monitored 6–8 weeks after treatment initiation to achieve stable levels, and levothyroxine should be titrated to normalize TSH levels. In central hypothyroidism, free T4 should be monitored instead of TSH. Additionally, adrenal insufficiency should be ruled out and treated prior to initiating thyroid hormone replacement to prevent adrenal crisis(10).
Routine monitoring for secondary adrenal insufficiency in ICI-treated patients is controversial. Morning cortisol levels with or without ACTH may be obtained in patients receiving anti-CTLA-4 inhibitors or combination therapy but are not typically recommended for anti-PD-1/PD-L1 monotherapy without symptoms. Adrenal insufficiency is treated with glucocorticoid replacement such as hydrocortisone or prednisone, with education on sick day dosing to prevent adrenal crisis. Stress dosing and hospitalization for IV hydration or parenteral glucocorticoid therapy are necessary for severe illness or procedures. If central hypothyroidism is diagnosed, glucocorticoids should be administered before thyroid hormone replacement to avoid adrenal crisis. High-dose glucocorticoids may be required for severe hypophysitis-related symptoms like headaches and vision changes. Immunotherapy can generally continue with proper hormone replacement therapy once severe symptoms resolve(11).
Primary adrenal insufficiency (PAI), if untreated, can be life-threatening and requires prompt glucocorticoid replacement therapy. The dose and route should be based on clinical status, with lifelong glucocorticoid and mineralocorticoid replacement therapy required to prevent hypotension and hyperkalemia. ICI therapy does not need to be discontinued unless symptoms are severe(12).
ICI-related diabetes mellitus (ICI-DM) often presents as diabetic ketoacidosis (DKA) in 60–85% of cases. Immediate treatment follows standard DKA protocols, including intravenous insulin, fluid resuscitation, and correction of electrolyte abnormalities. Patients require permanent insulin therapy, managed similarly to classic type 1 diabetes. High-dose glucocorticoids are not recommended as they do not improve outcomes and may exacerbate hyperglycemia. Experimental therapies, such as mesenchymal stem cell treatment, have shown potential in preclinical models, but further research is needed(13, 14).
Of the 31 patients with hyperthyroidism, 4.3% received antithyroid medication, while the majority were monitored. Most subsequently developed hypothyroidism. Immunotherapy was temporarily paused in 5 cases, but no treatment discontinuation was required. Treatment of ICI-induced thyrotoxicosis due to destructive thyroiditis is mainly supportive, with beta-blockers used for symptom control. In cases of ICI-induced Graves’ disease, anti-thyroidal medications, radioactive iodine, or surgery may be considered based on patient preference and clinical presentation. High-dose glucocorticoids are not recommended for thyroid-related immune-related endocrine events (irEEs) as they do not alter disease progression.
A total of 91 patients developed hypothyroidism, with thyroid autoantibodies detected in 31.9%. Among these, 34.4% were anti-TPO positive, 13.7% were anti-Tg positive, 17.2% were positive for both, and 27.5% were negative for both. Thyroiditis was observed in 86.2% of patients with available thyroid ultrasound data. Thirty-one patients developed hyperthyroidism. Thyroid autoantibodies were detected in 58% of these patients, primarily anti-TPO. Thyroiditis was present in 75% of patients with available ultrasound data.
Hypophysitis occurred in 12 patients, all of whom required hormone replacement therapy. One patient received high-dose steroids. Immunotherapy was temporarily paused in seven patients. Diabetes mellitus developed in one patient, who was treated with insulin therapy and had a temporary pause in immunotherapy. Adrenal insufficiency occurred in 19 patients, all of whom received hormone replacement therapy. Nine patients received high-dose corticosteroids. Immunotherapy was paused in two patients and discontinued in three.
For male hypogonadotropic hypogonadism, testosterone replacement can be initiated if indicated and there is no contraindication. In cases of ICI-CDI, desmopressin can be used to manage extreme thirst and polyuria, with temporary suspension of ICI therapy in severe dehydration and hypernatremia until stabilization is achieved.’’

Reviewer 2 Report
Comments and Suggestions for Authors
The article provides a comprehensive analysis of endocrine-related adverse events (irAEs) in cancer patients undergoing immune checkpoint inhibitor (ICI) therapy. The retrospective study, which includes 139 patients, explores the incidence, timing, management, and outcomes of these side effects. While the study is valuable, there are areas where the analysis could be deepened and clarified.
-
- I suggest that the authors provide a more detailed discussion of the biological mechanisms underlying ICI-induced endocrine irAEs. Specifically, the potential immunological pathways leading to organ-specific autoimmunity could be elaborated upon. For example, exploring how ICIs may disrupt immune tolerance or promote cross-reactivity with endocrine organs would strengthen the discussion.
- Including relevant literature to support this aspect could add depth. The authors might consider referencing studies on autoimmune processes triggered by ICIs to offer readers a broader context.
-
Role of Micronutrients:
- I encourage the authors to consider the role of micronutrients, such as zinc, selenium, and iron, in supporting the immune system and potentially modulating the severity of irAEs. This would broaden the scope of the discussion and make the findings more relevant to clinical practice.
- To aid in this, I suggest incorporating the findings from the article DOI: 10.3390/nu15112611, which discusses the influence of micronutrients on immune responses. This addition could enrich the discussion on supportive strategies for managing irAEs.
-
- The presentation of tables, such as "Baseline Characteristics," could be improved. I recommend reorganizing the data to enhance clarity and readability, for instance, by condensing the information into more concise formats or restructuring redundant elements.
- The manuscript contains some minor language and formatting issues, which I suggest addressing:
- Line 99-100: "the least common was diabetes mellitus (0,7%)" — Replace the comma with a period to conform to English decimal formatting ("0.7%").
- Line 114: "median latency of 4,5 months" — Correct to "4.5 months."
- Line 31: "improved clinical outcomes. How- ever" — This hyphenation should be removed; "However" should be written as a single word.
The authors are to be commended for their thorough and relevant study of endocrine irAEs in ICI therapy. However, I believe the inclusion of the following elements would strengthen the article:
- A deeper exploration of the mechanisms underlying endocrine irAEs.
- A discussion on how nutritional and micronutrient factors might influence immune-related outcomes.
- Improved presentation of data to enhance clarity for readers.
I encourage the authors to consider these suggestions in future revisions. The addition of reference DOI: 10.3390/nu15112611 would also add valuable insights to the discussion on immune support strategies.
Author Response
In response to the reviewers’ feedback, we have carefully addressed all concerns and incorporated the recommended changes. Below, you will find a detailed response to each reviewer, outlining the revisions made to the manuscript. Key updates include:
- Comprehensive specification of the study design and inclusion/exclusion criteria.
- Detailed statistical methods and analyses used in the study.
- Addressing the limitations due to sample size and retrospective nature.
- Expanding the discussion to include biological mechanisms, the role of micronutrients, and additional context supported by relevant citations.
- Updates to disease staging and integration of recommended references.
We believe these revisions strengthen our manuscript and enhance its contribution to the understanding of immune-related endocrine adverse events. We hope this revised version meets the expectations of the reviewers and the editorial team. Thank you for the opportunity to improve our work, and we look forward to your feedback. The revisions made in response to the reviewers' comments are detailed below, and the corresponding changes have been highlighted in yellow within the main manuscript.
1- We have expanded the discussion to include a more detailed examination of the biological mechanisms underlying ICI-induced endocrine irAEs. Specifically, we have elaborated on the potential immunological pathways leading to organ-specific autoimmunity. This includes an exploration of how ICIs may disrupt immune tolerance and promote cross-reactivity with endocrine organs.
Additionally, we have integrated relevant literature on autoimmune processes triggered by ICIs to provide readers with a broader context. We believe these additions strengthen the discussion and address your insightful comments.
2- Thank you for your insightful suggestion. We have addressed the role of micronutrients, including zinc, selenium, and iron, in supporting the immune system and their potential influence on the severity of irAEs. This addition broadens the scope of our discussion and enhances its relevance to clinical practice.
Moreover, we have specifically incorporated findings from the article DOI: 10.3390/nu15112611, as recommended. This reference has provided valuable insights into the influence of micronutrients on immune responses, enriching our discussion on supportive strategies for managing irAEs.
The detailed changes reflecting these additions can be found below.
‘’Nutrients play a pivotal role in modulating immune function and maintaining immune homeostasis. Macronutrients, such as proteins, carbohydrates, and fatty acids, along with micronutrients like vitamins, minerals, antioxidants, and probiotics, influence both innate and adaptive immunity. They regulate inflammation and cytokine expression, modulate immune cell signaling, and impact T and B cell activation, differentiation, and antibody production. These interactions underscore the importance of optimizing nutritional strategies to support immune function and potentially mitigate immune-related adverse events in patients undergoing immune checkpoint inhibitor therapy(36). Recent studies have found that high serum levels of selenium and zinc are associated with better prognosis in cancer patients, while elevated copper levels are linked to poorer survival outcomes. These associations suggest that serum levels of certain elements may serve as prognostic markers across various cancers(38).’’
3- Thank you for your feedback. The suggested language and formatting corrections have been addressed in the revised manuscript. We appreciate your attention to detail.

Reviewer 3 Report
Comments and Suggestions for Authors
interesting paper whose aim is the relationship between immune check-point inhibitor drugs and the endocrine dysfunctions induced by them. With the "curious" best effect in patients in whom the greatest endocrinological dysfunctions are created. Retrospective paper rather recent, from whose introduction we know which are the treated pathologies, solid neoplasms and the drugs that were used and in which line. It is advisable, given the table, to be more explicit on the stage of the disease in which the immune check-point inhibitors were introduced, perhaps with a second table. The materials and methods are descriptive on the various deficiencies found and on the measures taken for the necessary corrections. The discussion is intended to be a wake-up call for colleagues who treat patients with these drugs. Very interesting is the conclusion on patients who do not develop endocrinological deficiencies and who at the same time do not respond or do so less to immune check-point inhibitor drugs. Only a high volume center as we described in our previous paper can come into contact with so many pathologies related to the use of these drugs (doi.org/10.3390/jcm12072708 to be cited in the bibliography) iconography to be implemented, good English, the bibliography supports the paper
Author Response
In response to the reviewers’ feedback, we have carefully addressed all concerns and incorporated the recommended changes. Below, you will find a detailed response to each reviewer, outlining the revisions made to the manuscript. Key updates include:
- Comprehensive specification of the study design and inclusion/exclusion criteria.
- Detailed statistical methods and analyses used in the study.
- Addressing the limitations due to sample size and retrospective nature.
- Expanding the discussion to include biological mechanisms, the role of micronutrients, and additional context supported by relevant citations.
- Updates to disease staging and integration of recommended references.
We believe these revisions strengthen our manuscript and enhance its contribution to the understanding of immune-related endocrine adverse events. We hope this revised version meets the expectations of the reviewers and the editorial team. Thank you for the opportunity to improve our work, and we look forward to your feedback. The revisions made in response to the reviewers' comments are detailed below, and the corresponding changes have been highlighted in yellow within the main manuscript.
1-The section on the stage of the disease has been updated, and the following information has been added.
‘’ Eighty percent of the patients included in the study had metastatic disease, 17% had locally advanced disease, and 2% were in the early stage’’.
2- Thank you for your insightful suggestion. We are pleased to inform you that we have included a citation to the article 'doi.org/10.3390/jcm12072708' as recommended. This reference has been integrated into the manuscript to support the discussion and provide additional context to the findings.
